# Radiomic Features Prognosticate Treatment Response in CAR-T Cell Therapy

**DOI:** 10.3390/cancers17111832

**Published:** 2025-05-30

**Authors:** Yoganand Balagurunathan, Jung W. Choi, Zachary Thompson, Michael Jain, Frederick L. Locke

**Affiliations:** 1Department of Machine Learning, H Lee Moffitt Cancer Center, Tampa, FL 33612, USA; 2Department of Diagnostic & Interventional Radiology, H Lee Moffitt Cancer Center, Tampa, FL 33612, USA; jung.choi@moffitt.org; 3Department of Biostatistics & Bioinformatics, H Lee Moffitt Cancer Center, Tampa, FL 33612, USA; zachary.thompson@moffitt.org; 4Department of Blood and Marrow Transplant, H Lee Moffitt Cancer Center, Tampa, FL 33612, USA; michael.jain@moffitt.org

**Keywords:** image biomarkers in lymphoma, radiomics in CAR-T, response to CAR-T, axi-cel biomarkers

## Abstract

We propose using quantitative imaging metrics (radiomics) derived from the standard of care patient imaging (positron emission tomography/computed tomography, PET/CT) to prognosticate response to advanced axicabtagene ciloleucel (axi-cel) cellular therapy in patients treated for refractory/relapsed Diffused large B-cell lymphoma (DLBCL). We find radiomic (shape, size) characteristics on the imaging scans (PET/CT) of extra nodal lesions are prognostic of survival outcome. These metrics have a high translational ability and can be used for patient selection that can benefit and spare others from these advanced treatments.

## 1. Introduction

Lymphomas, heterogeneously diverse hematological malignancies that arises from B cells, T cells, or natural killer (NK) cells, are broadly classified into Hodgkin lymphoma (HL) and non-Hodgkin lymphoma (NHL). Each disease presents unique clinical and genetic features requiring tailored treatments [1,2]. Diffuse Large B-Cell Lymphomas (DLBCLs) are the most common form of non-Hodgkin lymphoma, accounting for about 30% of the cases, and are aggressive disease subtypes with a 5-year survival of 65% across all grades [3,4]. It is reported that about 30 to 40% of cases may present with advanced relapsed/refractory (R/R) stages that currently rely on salvage treatment that includes intensive chemotherapy and a stem cell transplantation [5]. Axicabtagene ciloleucel (axi-cel), chimeric antigen receptor (CAR) T-cell therapy using a CD19 target, has demonstrated superior levels of durable response in R/R DLBCL, currently recommended as a second-line treatment [6,7,8]. There have been efforts to develop clinical biomarkers, such as the International Prognostic Index (IPI), to prognosticate survival across risk groups in DLBCL based on retrospective data [9]. This index uses pretreatment clinical variables such as serum Lactate Dehydrogenase (LDH) and the Eastern Cooperative Oncology Group (ECOG) status, clinical stage, and extranodal site, which have been shown to be useful at diagnosis of the disease [10]. These markers do not provide a clear benefit in predicting disease progression or relapse beyond disease assessment at baseline [11]. Current clinical consensus criteria for disease staging and response include the Cheson and Lugano classification [12,13], but they are unable to prognosticate treatment response. Metabolic tumor volume (MTV) measure gross tumor burden, and these metrics have been recently shown to be prognostic of treatment response in CAR-T [14,15,16]. However, these metrics are time-consuming to compute and do not provide any insights into the disease progression.

There is an urgent clinical need to develop a biomarker that can be used to identify patients who would benefit from cellular therapy. Improvements in imaging modalities have led to better staging and detection of DLBCL on ^18^F fluorodeoxyglucose (^18^F-FDG) positron emission tomography (PET), along with high-resolution computed tomography (CT) imaging that has allowed for more precise evaluation of disease condition, tumor biology, and its microenvironment [17].

Quantitative imaging metrics (radiomics) have shown enormous promise in reflecting the pathophysiology of the tumor and describing the textural heterogeneity and shape characteristics, and have been shown to prognosticate the disease progression across oncological diseases [18,19,20]. These metrics have been recently shown to be useful in predicting outcomes in DLBCL [21,22]. In this study, we propose to use radiomic features on patient imaging (PET, CT) of DLBCL and test their ability to prognosticate treatment response for CAR-T therapy, complementing our predictive models presented in our previous work [22]. Our study overview is illustrated in Figure 1.

## 2. Materials and Methods

### 2.1. Patient Data

The study retrospectively curated 155 patient records with R/R diffuse large B-cell lymphoma (DLBCL) who had received CAR-T therapy (axi-cel). We obtained 100 patients from the H Lee Moffitt Cancer Center, and 55 anonymized patient records were obtained from the clinical trial consortium (Zuma-1, sponsor Kite Pharma). Patients with non-measurable lesions or with or without baseline PET imaging were excluded from the study. Bridging therapy was defined as any lymphoma-specific therapy given before CAR T-cell infusion, prior to the start of fludarabine cyclophosphamide chemotherapy for lymphodepletion, previously presented [14,22].

Our study was approved by the Institutional Review Board (IRB) at the University of South Florida /Moffitt Cancer Center. Patients with baseline imaging (^18^F FDG-PET/CT) prior to CAR T-cell therapy were included. Most patients received bridging therapy as a standard of care before treatment, defined as any lymphoma-specific therapy. Table 1 shows the patient cohort characteristics. Imaging data corresponding to whole-body CT and FDG-PET (attenuated corrected) image voxel values were converted to standardized uptake value (SUV) reference before our analysis.

### 2.2. Metabolic Tumor Volume

We used custom tools implemented on MIM PACS (version 6.8.4, MIM Software^®^, Cleveland, OH) to semi-automatically identify lesions with SUV uptake over a reference liver region (≥2 cm diameter), which was manually located, following the PERSIST criteria [23]. A clinical expert (radiologist J.W.C.) evaluated these abnormal regions and removed falsely detected regions. Most common false detections were due to physiological processes (brain/bladder/etc.), and a few others were removed due to inflammation. We followed the consensus criteria for lesion identification on the PET image scans [23,24], and considered voxels over the 41% of SUVmax to define the lesion boundary on these images. We then summed metabolically active regions across the body to obtain Metabolic Tumor Volume (MTV), reported in multiple scales, milliliters (mL), and cubic centimeters (cc). The metrics have been previously presented in DLBCL patients [14].

### 2.3. Radiological Review

Lesions were semi-automatically identified on the PET scan and reviewed by our research/clinical radiologist (J.W.C). Additional details about the lesion’s anatomical location and association with the lymphatics (nodal vs. extranodal) and volume/size were recorded. There were 1058 lesions reviewed across the patient cohort, of which 342 were related to nodal (lymphatics), and 616 were related to extranodal (non-lymphatic). The patient sub-cohort was organized by grouping the largest lesion in the nodal (*n* = 124) and extranodal (*n* = 94), independently. The nodal cohort had most lesions in the abdomen, pelvis, and neck, which are three major organ sites. The extranodal group had the most lesions in the lungs, bone, and liver, which were the major organ sites across the patients in the cohort.

### 2.4. Radiomics

The CT and FDG-PET imaging were resampled to a common reference resolution of 1 × 1 × 1 mm^3^ using bilinear interpolation. The PET images were standardized to SUV units using the activity concentration to the dosage of ^18^F-FDG injected volume and patient body weight. The abnormal region was converged on the PET images and was translated to CT. We extracted 306 radiomic features for the identified lesion in each modality: CT and PET (SUV) images, totaling 612 features for a lesion. We categorized imaging radiomic features into three broad functional categories: Size (*n* = 38), Shape (*n* =9), and Texture (*n* = 259); details on the feature descriptors are deferred to Appendix A, which were previously reported [22]. The radiomic feature descriptor definition followed the recommendations of the Image Biomarker Standardization Initiative (IBSI) consensus criteria [25,26,27]. Features with a minimal change (coefficient of variance ≤ 3%) across the patient samples were removed due to the invariant nature of the metric.

### 2.5. Statistical Analysis

The relationship between the radiomics feature-based principal components (PC) and the metabolic tumor volume (MTV) was assessed using Spearman’s correlation coefficients (see Table 2). We used Kaplan–Meier (KM) survival analysis to estimate the survival function of the identified patient groups, separated using the imaging metrics (radiomics). We used the midpoint (Median) on the PC metric to divide patients into groups and evaluate the treatment response measured by survival time to event (OS/PFS). The process was independently tested across the feature categories (size, shape, texture). We quantified the difference in the survival functions using log-rank statistical hypothesis testing to estimate the significance [28]. We computed the proportional hazard ratio (HR) for these cohorts using the Cox-regression model [29] and reported concordance (C-index), to continuous event time measured by survival time (OS/PFS) [30]. A *p*-value of less than 0.05 was considered statistically significant in our analysis. We corrected for multiple testing by computing adjusted *p*-value (q-value or false discovery rate) [31].

## 3. Results

The study assessed patients’ baseline PET and CT scans (*n* = 155), and we formed sub-cohorts based on the anatomical location of the largest lesion related to the lymphatic or nodal (*n* = 124) and non-lymphatic or extranodal (*n* = 94). The most frequently involved nodal sites (lymphatic) were the abdominal (46.77%), pelvic (17.74%), and mediastinal (8.87%). The most frequently involved extranodal sites were lung (36.17%), musculoskeletal (21.28%), and pelvic (8.51%). The radiomic features were categorized into functional categories: Size (*n* = 38), Shape (*n* = 9), and Texture (*n* = 259), and the lesions were characterized independently across the imaging modalities (CT and PET).

We obtained principal components (PC) on features in each of the feature categories. We found the size-based PCs in extranodal CT and PET (-SUV) images showed the highest correlation with metabolic tumor volume (MTV), ρ = −0.451 and 0.449, respectively. Texture and Shape-based PCs on lymphatic lesions in PET (SUV) scans had the lowest correlation with MTV, a ρ = 0.175 and 0.271, respectively; see Table 2. We report the top five individual features in each of the functional categories, selected based on those with the highest loading factors for the respective principal components. In the shape category, the shape-PCs are related to compactness and sphericity-based features (loading of −0.43). We found that size-based PCs showed higher loading related to volume fraction (at 10% and 90%), which are extracted in PET images. The texture-based PCs showed higher loading factors related to co-occurrence type features in PET images (see Table 3).

We tested the ability of radiomics-based principal components to predict treatment outcome, measured by the overall survival, OS, and progression-free survival, PFS (censored at 1 year). We then evaluated shape, size, and texture-based radiomic PCs derived from PET and CT image data in the extranodal cohort that were significant (*p* < 0.05) for OS and PFS.

We found that the shape- and size-based radiomic PC metrics on extranodal cohort were significant (*p* < 0.05) for OS and PFS, in PET images (see Table 4). We illustrate patients’ image scans with high and low shape metrics (see Figure 2). We used the Cox-regression model to assess patients’ proportional hazard (HR) to treatment outcome, measured by survival (OS/PFS). We first evaluated the clinical metric, MTV, to outcome, and found an HR of 3.885 and 3.64 for nodal (lymphatic) and extra-nodal (non-lymphatic) cohorts. The c-index is 0.67/0.65 for nodal (lymphatic)/extra-nodal (non-lymphatic) cohorts (see Table 5). We found that size and shape PCs on extra nodal CT and PET radiomic PCs were significant and showed higher risk for adverse outcomes (c-index 0.59 to 0.6 for shape and 0.61 to 0.6 for size, in OS/PFS, respectively). Texture-based radiomic PCs on extranodal CT show significance (c-index 0.61/0.6 for OS) (see Table 5).

We compared the radiomic feature-based principal component (PC1) computed in each of the feature categories (size, shape, texture) across patients separated based on their clinical disease condition at follow-up time (disease progression and non-progression). We found a significant difference in the radiomics PC metric across all feature categories (see Figure 3a). We then repeated the comparison in patients with lower tumor volume (MTV < 147.5) and higher tumor volume (MTV ≥ 147.5). The MTV cut-point of 147.5 mL was established in previously published findings [14], to prognosticate patient response. We found that shape PC and size PC metrics show significant differences between the progressors and non-progressors in the lower tumor volume group (see Figure 3b). In the larger tumor volume groups, we do not see significant differences in clinical progression (See Figure 3c). We used the radiomic PCs to prognosticate treatment outcomes (OS/PFS), using the Kaplan–Meier survival analysis across, and statistical significance was assessed using a log-rank test (See Figure 4).

## 4. Discussion

This study used retrospective patient data to develop a quantitative imaging metric (radiomics) signature derived from their baseline PET/CT scans that could be used as a biomarker to prognosticate response (1-year) in patients undergoing axi-cel therapy. We systematically assessed the lesions based on their role in the lymphatic system (nodal, extra-nodal) and categorized them based on the functional description (size, shape, texture). We showed that radiomic-based features (PC on shape and size) derived from the extra-nodal lesions have a higher risk of adverse outcomes and are a prognostic indicator of survival.

The heterogeneous nature of lymphoma disease makes it challenging to assess and prognosticate treatment response [32,33]. Current clinical consensus assessment criteria [12,13], do not fully capture the disease condition, nor can they predict disease prognosis, especially in axi-cel therapy, leaving a need for better biomarkers. Characterizing subtle physiological changes seen on radiological imaging with quantitative metrics (radiomics) that describe them has been shown to be related to disease conditions and their response outcomes [18,19,20]. This study is unique in many aspects; firstly, we use lesion-based metrics across many organ sites and form predictors to evaluate the ability to prognosticate survival outcome. Secondly, we subdivided the lesions at the patient level based on nodal (or lymphatic) and extra-nodal (or non-lymphatic) sites. Our study showed that the principal components of radiomic features have the ability to prognosticate patient treatment outcomes to axi-cel treatment. Few other teams have shown similar outcomes without distinguishing the disease sites [21,34].

Beyond tumor metabolic activity, the shape of the lesions observed on imaging scans (PET, CT) can potentially play a diagnostic role in lymphoma [25,35,36,37,38]. In current clinical practice, lesion tracer uptake (PET activity) is assessed on a point scale with qualitative reference to the normal metabolic uptake (liver or mediastinum) [12,39]. Grayscale characteristics in PET/CT image scans can be quantified with metrics describing the lesions’ size, shape, or texture. There has been abundant literature that has shown the usefulness of radiomic features in predicting the pathological cancer status, oncogenic mutations, or clinical response [18,40,41,42,43]. There have been many studies that have attempted to assess the response of axi-cel therapy in DLBCL [7,44]; most useful quantitative biomarkers have been limited to gross measures such as the MTV. These metrics are known to describe the extent of the disease and have been reported to be prognostic of treatment response [14,45,46,47]. In our prior work [22], we tested the ability of the radiomic metrics to predict the treatment outcome of axi-cel therapy (AUC 0.68, compared to TMV 0.72), while others have shown the utility of radiomic features (on PET images) related to maximum intensity, skewness, major axis, and gray-level emphasis, which were predictive of the treatment outcome (AUC of 0.73, compared to TMV of 0.66) and prognostic of OS/PFS for 3-month follow up [21]. In comparison, our study identifies the largest lesion in nodal/extra nodal and characterizes them using descriptors in three major feature categories (size, shape, texture), and tests their ability to prognosticate treatment outcome (OS/PFS). The study shows that shape-based descriptors in the extra-nodal cohort on PET images show a significant survival difference (OS/PFS) after axi-cel treatment (1-year). Our findings have been validated by several groups, and patients with extra-nodal lesions are reported to have worse outcomes [48,49]. In a multivariate analysis, the number of sites of extra nodal disease with initial high LDH at lymphodepletion was reported to have an inferior survival outcome [49]. In lymphoma, single variable-based models with reproducible thresholds are often used to prognosticate patient outcomes [50,51]. Our methodology uses PCs on the radiomic features in functional categories, where the size category has 38 features, the shape category has nine features, and the texture category has 259 features, see Appendix A. An ensemble combination (like principal component, PC) of these features with similar function, in these categories, provides a better basis that could be used as a biomarker to prognosticate treatment response. In our study, we found non-size-based PCs (Shape-based PC1) on extranodal lesions to be prognostic (*p* < 0.05) of OS/PFS on PET, and non-size-based PCs (Shape and Texture based PC1) on CT images were also found to be prognostic (*p* < 0.05) of OS/PFS (see Table 4, Figure 4) and higher risk for survival outcome (see Table 5). We find these radiomic PCs (Shape, Texture and Size) had a low level of correlation with the metabolic tumor volume metric (MTV), computed at the patient level (highest Spearman’s, ρ of 0.45, 0.37, 0.55, respectively). In solid tumors, it has been reported that most proliferative lesions are non-circular, and irregular compared to benign lesions [37,52,53]. A recent publication [54] reported that a high metabolic heterogeneity measured as a cumulative SUV-histogram on PET images has a poor prognosis for patients with DLBCL.

There are few studies that have tracked the toxicities related to advanced immunotherapy [55], which can certainly be an investigative area. Recent advancements in genomics biomarkers in other blood-based malignancies have shown promise in prognosticating disease progression [56,57]. Advancements in artificial intelligence methods will certainly play a role in identifying finer patterns in patients that could be used to improve patient care in these advanced treatments [58,59,60], there are several data requirements that need to be considered prior to clinical utility [61,62].

Our study demonstrates the usefulness of lesion-based radiomic PCs to prognosticate treatment outcomes. This study has identified shape and size characteristics (shape PC and size PC, texture in CT) on imaging scans that can serve as surrogates to treatment response. The study contributions can be summarized as follows: (a) identifying the role of shape/size-based features, (b) association with extra nodal lesions.

These metrics, independently or collectively, could help select patients who would benefit from advanced axi-cel therapy and improve care delivery effectively. Our study, despite showing novel findings, would require secondary validation for clinical translation.

## 5. Limitations

Our study is limited by retrospective data collection, and our findings would need an independent validation cohort prior to clinical translation. Our study did not collect or relate to cell therapy-related toxicities like cytokine release syndrome (CRS), immune effector cell-associated neurotoxicity (ICANS), prolonged cytopenia, tumor lysis syndrome, and others as previously reported [55].

## 6. Conclusions

The study investigated the prognostic role of radiological image metrics (radiomics) on PET/CT images and related them to responses to immunotherapy. The study identified radiomic signatures based on shape and size features to be prognostic of axi-cel therapy. These image-based metrics derived at the lesion level will potentially allow greater clinical translation in recruiting patients who will benefit from these treatments.

## Figures and Tables

**Figure 1 cancers-17-01832-f001:**
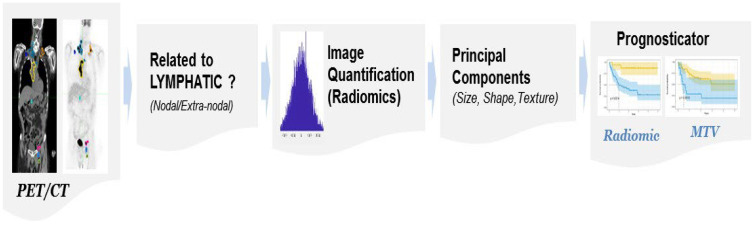
The study process flow shows the use of patient-level imaging metrics (radiomics, MTV).

**Figure 2 cancers-17-01832-f002:**
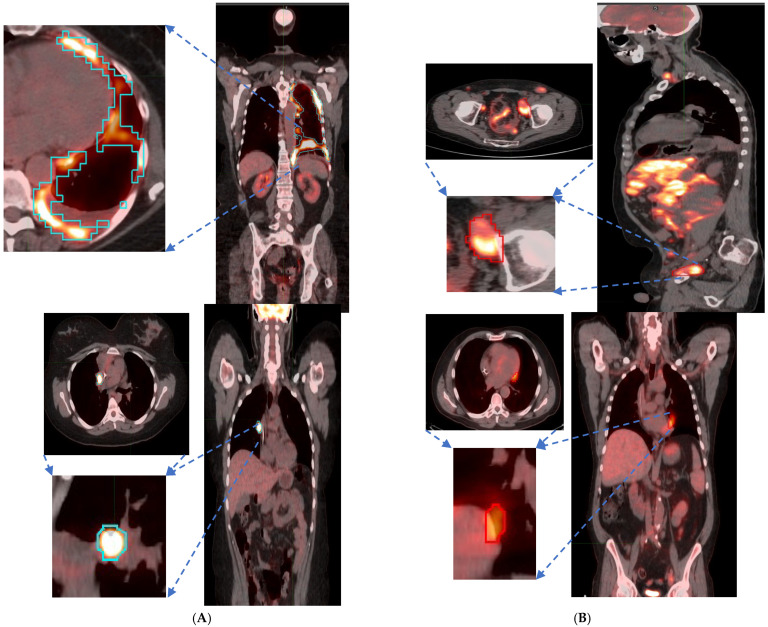
Patient scans showed representative slices in the fused image (CT/PET) with a pointing arrow towards lesions with significant uptake. Representative patients (Lung and Abdominal) image slice for (**A**) High shape metric and (**B**) Low shape metric.

**Figure 3 cancers-17-01832-f003:**
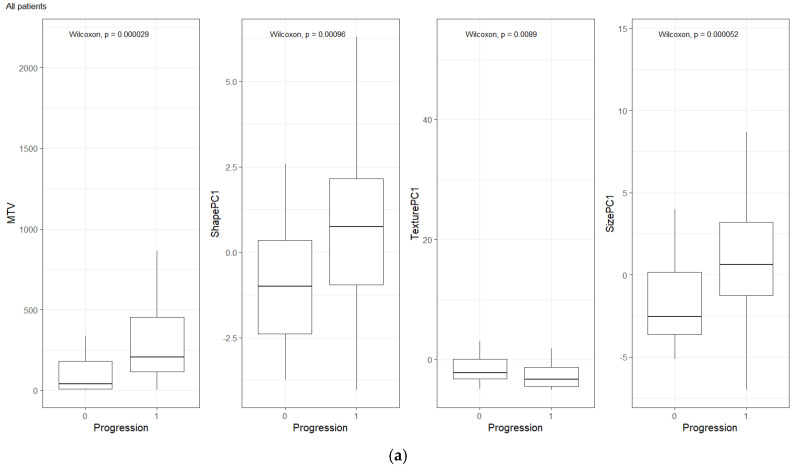
Distribution of patients based on clinical outcome (progression at 1-year) for MTV, Shape PC and Texture PC for (**a**) all patients, (**b**) patients with lower total tumor burden, and (**c**) patients with higher tumor burden. The cut-point value was based on prior work (*Blood Adv.*, 4(14) 2020) [14].

**Figure 4 cancers-17-01832-f004:**
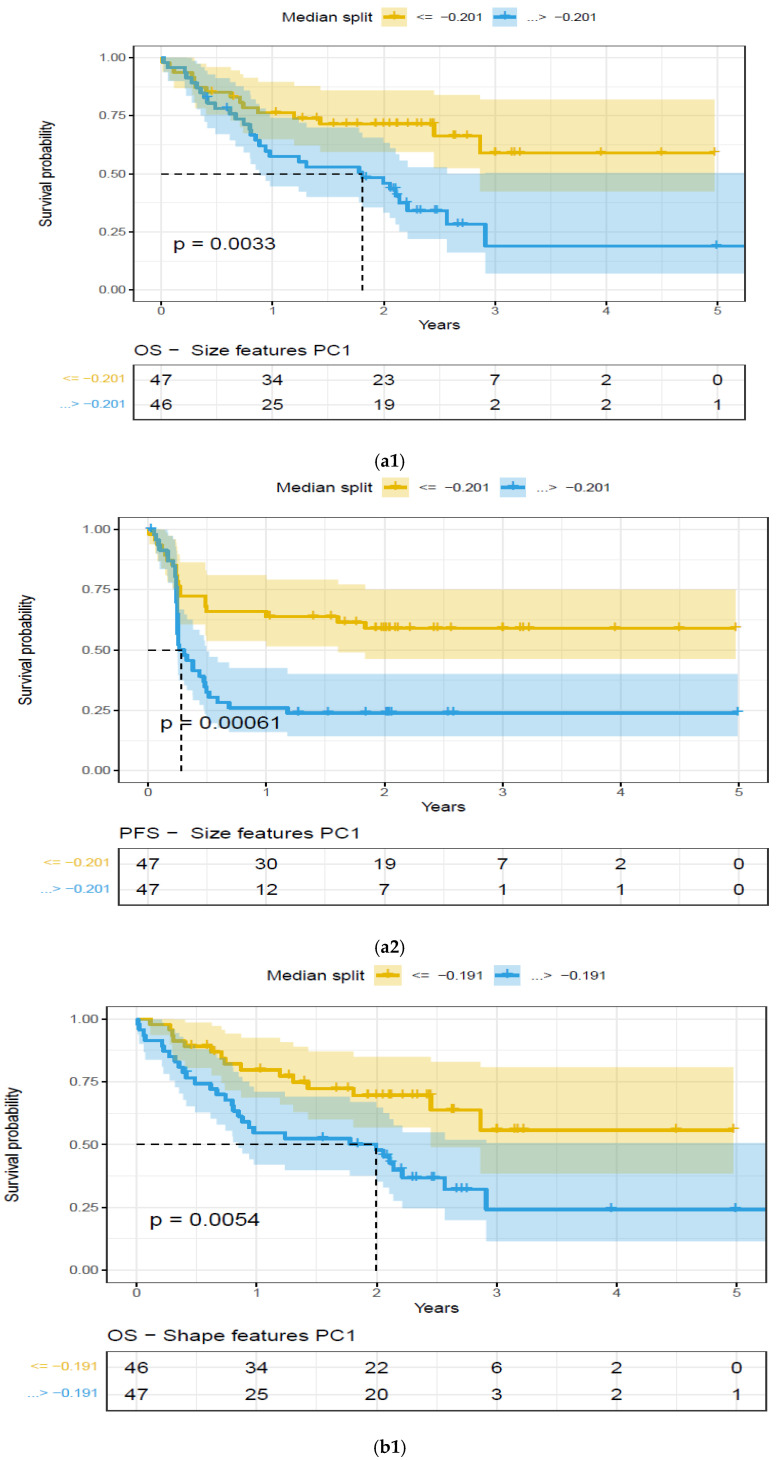
Kaplan–Meier (KM) plots of patient cohort separated using median split based on features extracted on the largest extra-nodal lesions in a patient’s PET/SUV scans, for feature categories based on Principal Component (PC1)—Overall survival. Feature categories PC1, KM plots, (**a**) size-based principal components (PC1)—OS (**a1**) and PFS (**a2**), (**b**) shape-based principal components (PC1)—OS (**b1**) and PFS (**b2**), (**c**) metabolic tumor volume—OS (**c1**) and PFS (**c2**), (See details in Table 4).

**Table 1 cancers-17-01832-t001:** Patient demographics for the data used for radiomic analysis.

Characteristics	All Patients * (N = 155)	Subcohorts (Lesion Level)	
		Extra-Nodal (*n* = 94)	Lymphatic (*n* = 124)
Age (mean, median, std.dev)	60.1 (63, 12.2)	59.4 (63.5, 12.8)	61 (63, 10.9)
Sex (male/female/unavailable)	61/39/55	36/22	53/29
LDH (mean, median, std.dev)	400.5 (266, 348.25)	448.3 (275.5, 406.79)	408.6 (267.5, 353.4)
ECOG			
0–1	83	48	66
2–3	17	10	16
One Year Progression	/death No Yes Unavailable 55	24 (41.4%) 34 (58.6%)	38(46.3%) 44 (53.7%)
Stage			
I/II III/IV	22 78	10 48	14 68
Bridge Therapy			
Yes No	Yes: 50 No: 50	Yes: 28 No: 30	Yes: 41 No: 41
Unavailable	55		
Axi-cel therapy			
Trial (cancer center) Consortium (Zuma-1)	100 55	58 36	82 42

* Some clinical variables were unavailable for consortium patients. Lactate Dehydrogenase (LDH), Eastern Cooperative Oncology Group (ECOG), axicabtagene ciloleucel (axi-cel).

**Table 2 cancers-17-01832-t002:** The relationship between metabolic tumor volume (MTV) and radiomics features PC (principal components) across feature categories was assessed using Spearman’s correlation coefficient.

Radiomic Metric (Principal Component, PC)	Spearman Correlation (ρ)	*p*-Value
Lymphatics (CT Images)		
Size PC1	0.368	0.0000268
Shape PC1	0.3002	0.0007
Texture PC1	−0.345	0.00008
Extra-Nodal (CT Images)		
Size PC1	−0.4519	0.000004
Shape PC1	0.3698	0.00024
Texture PC1	0.5535	0.00
Lymphatics (PET Images)		
Size PC1	0.3698	0.000023
Shape PC1	0.2717	0.0023
Texture PC1	0.175	0.0518
Extra-Nodal (PET Images)		
Size PC1	0.4496	0.0535
Shape PC1	0.3590	0.00038
Texture PC1	−0.275	0.0073

**Table 3 cancers-17-01832-t003:** Top loading factors (absolute value) for principal components (PC1 to PC3) on individual radiomic features estimated in the entire cohort.

Features		Loading Factors			Median Value
		PC1	PC2	PC3	MTV (Median): 169.94 mL
Shape-Related Features					Shape-PC1 (Median): −0.1909
1	Compactness_1	−0.431	−0.122	0.024	
2	Compactness_2	−0.414	−0.074	0.025	
3	Spherical_disproportion	0.405	0.224	0.017	
4	Sphericity	−0.434	−0.141	0.020	
5	Asphericity	0.405	0.224	0.017	
Size-Related Features					Size-PC1 (Median): −0.2013
1	SUV (volume_at_intensity_fraction_10)	−0.031	−0.123	−0.338	
2	SUV (volume_at_intensity_fraction_90)	−0.147	−0.032	−0.149	
3	SUV (intensity_at_volume_fraction_10)	−0.097	−0.101	−0.297	
4	SUV (intensity_at_volume_fraction_90)	0.023	−0.129	−0.331	
5	SUV (volume_at_intensity_fraction_difference)	0.011	−0.121	−0.316	
Texture-Related Features					Texture-PC1 (Median): −2.9435
1	SUV (avg_coocurrence_joint_max)	0.030	0.001	−0.078	
2	SUV (avg_coocurrence_joint_average)	−0.012	0.054	0.003	
3	SUV (avg_coocurrence_joint_variance)	0.050	−0.104	0.020	
4	SUV (avg_coocurrence_joint_entropy)	−0.064	0.055	0.044	
5	SUV (avg_coocurrence_difference_average)	0.074	−0.129	0.060	

**Table 4 cancers-17-01832-t004:** Prognostic value of radiomics feature-based metrics (principal components, PCs), measured by overall survival (OS) and progression-free survival (PFS), 1 Year after axi-cel therapy. The lesions were stratified into sub-cohorts (nodal and extra-nodal) across modalities: (A) CT and (B) PET (SUV) imaging. The MTV (metabolic tumor volume) was computed at the patient level. Adjusted *p*-value (false discovery rate, q-value) was also reported; * *p* < 0.1.

**A. Metrics on Computed Tomography (CT) Imaging**					
	**Variable**	**Survival Statistics (Log-rank *p*-Value, q-Value)**			
		**OverAll Survival (OS)**		**Progression Free Survival (PFS)**	
		**Nodal (*n* = 126)**	**Extra Nodal (*n* = 94)**	**Nodal (*n* = 126)**	**Extra Nodal (*n* = 94)**
1	MTV (total body)	<0.0001 * (0.0003)	<0.0001 * (0.0002)	0.00024 * (0.0096)	0.00017 * (0.00096)
2	Shape-PC1	0.75 (0.75)	0.008 * (0.013)	0.67 (0.72)	0.017 * (0.0272)
3	Size-PC	0.18 (0.24)	0.0012 * (0.003)	0.22 (0.293)	0.00046 * (0.0012)
4	Texture-PC	0.58 (0.663)	0.0037 * (0.007)	0.72 (0.72)	0.0022 * (0.0044)
**B. Metrics on Positron Emission Tomography (PET) Imaging**					
	**Variable**	**Survival Statistics (Log-rank *p*-Value, adjusted *p*-Value)**			
		**Over All Survival (OS)**		**Progression free Survival (PFS)**	
		**Nodal (*n* = 126)**	**Extra Nodal (*n* = 94)**	**Nodal (*n* = 126)**	**Extra Nodal (*n* = 94)**
1	MTV (total body)	<0.0001 * (0.0003)	<0.0001 * (0.00069)	0.00024 * (0.00096)	0.00017 * (0.00096)
2	Shape-PC1	0.072 (0.1152)	0.0054 * (0.0108)	0.052 (0.0832)	0.0074 * (0.0148)
3	Size-PC	0.16 (0.2133)	0.0033 * (0.0088)	0.21 (0.2400)	0.00061 * (0.0016)
4	Texture-PC	0.37 (0.4229)	0.73 (0.730)	0.089 (0.11897)	0.43 (0.430)

**Table 5 cancers-17-01832-t005:** The proportional hazard ratio using the Cox regression model was used to assess the role of MTV and Radiomic features in treatment outcome (Overall survival, OS) and Progression-free survival, PFS). Evaluated across the sub-cohorts divided based on nodal status (nodal and extra-nodal) and modalities: (A) CT and (B) PET (SUV) related to OS, (C) CT and (D) PET (SUV) metrics related to PFS. Adjusted *p*-value (false discovery rate, q-value) was also reported.

**A. Metrics on CT Imaging: OS**					
	**Variable**	**Nodal (*n* = 126)**		**Extra Nodal (*n* = 94)**	
		**HR (*p*-Value, q-Value)**	**C-Index (CI)**	**HR (*p*-Value, q-Value)**	**C-Index (CI)**
1	MTV (total body)	3.885 (0.00001), 0.0001	0.67 [0.616,0.721]	3.644 (0.000093), 0.0004	0.65 [0.575, 0.721]
2	Shape-PC1	1.183 (0.5424), 0.5762	0.52 [0.426, 0.615]	2.383 (0.0062), 0.0123	0.6 [0.516, 0.69]
3	Size-PC	1.524 (0.1293), 0.172	0.56 [0.464, 0.647]	2.546 (0.0021), 0.0056	0.62 [0.54, 0.695]
4	Texture-PC	0.858 (0.5762), 0.5762	0.52 [0.427, 0.623]	2.227 (0.0103), 0.0164	0.59 [0.495, 0.682]
**B. Metrics on PET Imaging: OS**					
	**Variable**	**Nodal (*n* = 126)**		**Extra Nodal (*n* = 94)**	
		**HR (*p*-Value, q-value)**	**C-Index (CI)**	**HR (*p*-Value, q-Value)**	**C-Index (CI)**
1	MTV (total body)	3.885 (0.000014), 0.0001	0.67 [0.603, 0.733]	3.644 (0.000093), 0.0004	0.65 (0.57, 0.725)
2	Shape-PC1	1.488 (0.1518), 0.2023	0.55 [0.438, 0.653]	2.139 (0.0148), 0.0296	0.59 (0.504, 0.683)
3	Size-PC	1.542 (0.1188), 0.1901	0.56 [0.466, 0.65]	2.477 (0.0044), 0.0117	0.61 (0.516, 0.696)
4	Texture-PC	1.175 (0.5568), 0.6364	0.52 [0.426, 0.608]	0.899 (0.7275), 0.7275	0.51 (0.424, 0.602)
**C. Metrics on CT Imaging: PFS**					
	**Variable**	**Nodal (*n* = 126)**		**Extra Nodal (*n* = 94)**	
		**HR (*p*-Value, q-Value)**	**C-Index (CI)**	**HR (*p*-Value, q-Value)**	**C-Index (CI)**
1	MTV (total body)	2.486 (0.0004), 0.0015	0.62 [0.553, 0.68]	2.814 (0.0003), 0.0015	0.62 [0.565, 0.685]
2	Shape-PC1	1.111 (0.6683), 0.7241	0.51 [0.434, 0.586]	1.921 (0.0192), 0.0309	0.58 [0.494, 0.662]
3	Size-PC	1.355 (0.2194), 0.2926	0.54 [0.439, 0.639]	2.417 (0.0014), 0.0037	0.61 [0.519, 0.699]
4	Texture-PC	0.917 (0.7241), 0.724	0.52 [0.429, 0.611]	2.323 (0.003), 0.006	0.6 [0.51, 0.682]
**D. Metrics on PET Imaging: PFS**					
	**Variable**	**Nodal (*n* = 126)**		**Extra Nodal (*n* = 94)**	
		**HR (*p*-Value, q-Value)**	**C-Index (CI)**	**HR (*p*-Value, q-Value)**	**C-Index (CI)**
1	MTV (total body)	2.486 (0.0003), 0.0015	0.62 [0.563, 0.67]	2.814 (0.0003), 0.0015	0.62 [0.56, 0.689]
2	Shape-PC1	1.615 (0.0545), 0.0872	0.56 [0.463, 0.659]	2.078 (0.0087), 0.0175	0.6 [0.529, 0.665]
3	Size-PC	1.36 (0.2135), 0.2440	0.54 [0.447, 0.633]	2.598 (0.0009), 0.0025	0.6 [0.533, 0.674]
4	Texture-PC	1.521 (0.0910), 0.1214	0.56 [0.477, 0.638]	0.806 (0.4284), 0.4284	0.54 [0.424, 0.647]

## Data Availability

The de-identified patient records for those treated at Moffitt Cancer Center will be shared after the institutional data transfer agreement.

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
