# Peer review of "Radiomic Features Prognosticate Treatment Response in CAR-T Cell Therapy"

_cancers, 2025, doi:10.3390/cancers17111832_

Round 1
Reviewer 1 Report
Comments and Suggestions for Authors
This manuscript presents a retrospective radiomics analysis in a cohort of 155 patients with relapsed/refractory DLBCL treated with axi-cel. The study evaluates radiomic features derived from baseline PET/CT scans, categorized into shape, size, and texture metrics, and assesses their prognostic value for 1-year PFS and OS. Principal component analysis is applied to reduce dimensionality, and the association of radiomic components with clinical outcomes is compared to metabolic tumor volume.
1- While q values ​​are reported, there is no internal cross-validation, bootstrap, or external test set to assess the generalizability of the findings. Hazard ratios for PCs are modest, and C-indices range from 0.59–0.62, reflecting borderline discriminatory performance.
2- Prognostic value of PCs is presented in univariate Cox regression models. However, multivariate models are not presented to assess independence from known prognostic factors (e.g. IPI, ECOG, LDH).
3- Radiomic features are extracted at the lesion level, but survival outcomes (OS/PFS) are assessed at the patient level, leading to potential statistical inconsistency and oversimplification.
4- While radiomic PCs and MTV were compared descriptively, no formal statistical comparison of predictive models (e.g. ΔC-index, NRI, IDI) was performed.
5- The manuscript provides a minimal descriptive account of baseline variables, which undermines the robustness of the downstream survival analysis. Given the central role of patient heterogeneity in DLBCL outcomes, a more comparative analysis of baseline data is essential.
Author Response
Response to Reviewers:
Dear Editors, We thank the reviewers for their constructive critiques, which have greatly improved our work. We updated our manuscript to address the concerns.
Reviewer#1 (R#1):
This manuscript presents a retrospective radiomics analysis in a cohort of 155 patients with relapsed/refractory DLBCL treated with axi-cel. The study evaluates radiomic features derived from baseline PET/CT scans, categorized into shape, size, and texture metrics, and assesses their prognostic value for 1-year PFS and OS. Principal component analysis is applied to reduce dimensionality, and the association of radiomic components with clinical outcomes is compared to metabolic tumor volume.
R#1-Q#1
1- While q values ​​are reported, there is no internal cross-validation, bootstrap, or external test set to assess the generalizability of the findings. Hazard ratios for PCs are modest, and C-indices range from 0.59–0.62, reflecting borderline discriminatory performance.
Response to R#1-Q#1: We understand the concern. In our analysis, we compared our radiomic model (Principal components, PC on Size, Shape, Texture) to the clinical reference (tumor burden). We find the concordance index (C index) of our model (shape PC, non-size based) is comparable to metabolic tumor burden (MTB or MTV). Our findings are in line with prior reporting in Large B-cell Lymphoma (Jing et al 2023, EJNMMI 13 (92)).
The retrospective nature of the study limited the number of patients accrued for the study. The sample size requirement for the prognostic model limitation did not allow us to split the cohort for independent validation. As the reviewer correctly pointed out, the need for ensemble results. Prognostic modeling does not allow the use of cross-validation/bootstrap-type approaches.
Reference:
Jing et.al, EJNMMI Res. 2023 Oct 26;13:92. doi: 10.1186/s13550-023-01047-5
Zhou et.al, J Cancer Res Clin Oncol, 2023 Oct;149(13).
R#1-Q#2
2- Prognostic value of PCs is presented in univariate Cox regression models. However, multivariate models are not presented to assess independence from known prognostic factors (e.g. IPI, ECOG, LDH).
Response to R#1-Q#2.
We understand the concern. Elevated LDH and high IPI (international prognostic index) scores can suggest aggressive disease (as described previously). In this study, we show a univariate assessment for comparative purposes. Prior studies, including ours, have shown that clinical factors are marginal and do not significantly aid imaging metrics in predicting the risk of adverse outcomes. The MTV (metabolic tumor volume) has been shown to better prognosticator compared to clinical variables. Hence, we use MTV as a comparator in our study. We show the hazard ratio for mixed model variables (added to the supplemental section).
|
a1.Cox Model (OS) – MTV, Radiomics, Clinical - CT (Lymphatic) |
|||
|
|
Variables |
Hazard Ratio |
P-value |
|
|
MTV |
1.173 [1.067,1.29] |
0.00099* |
|
|
LDH |
1.00 [0.999,1.001] |
0.6704 |
|
|
Texture PC1 |
0.999 [0.932,1.07] |
0.96895 |
|
|
Shape PC1 |
1.018 [0.806,1.285] |
0.88256 |
|
|
Texture PC1: Shape PC1 |
0.994 [0.971,1.017] |
0.60156 |
|
b1.Cox Model (OS) – MTV, Radiomics & Clinical on PET Images (Lymphatics) |
|||
|
|
Variables |
Hazard Ratio |
P-value |
|
|
MTV |
1.202 [1.077,1.341] |
0.00104* |
|
|
LDH |
1.00 [0.999,1.001] |
0.59649 |
|
|
Texture PC1 |
1.024 [0.964,1.088] |
0.43701 |
|
|
Shape PC1 |
1.029 [0.86,1.231] |
0.75385 |
|
|
Texture PC1: Shape PC1 |
1.018 [0.995,1.041] |
0.12498 |
|
c1.Cox Model (OS) – MTV, Radiomics & Clinical - CT (Extra-nodal) |
|||
|
|
Variables |
Hazard Ratio |
P-value |
|
|
MTV |
1.142 [1.029,1.269] |
0.01291* |
|
|
LDH |
1.00 [0.999,1.002] |
0.54026 |
|
|
Texture PC1 |
0.999 [0.919,1.086] |
0.98241 |
|
|
Shape PC1 |
0.959 [0.689,1.335] |
0.80577 |
|
|
Texture PC1: Shape PC1 |
0.99 [0.961,1.02] |
0.50609 |
|
d1. Cox Model (OS) – MTV, Radiomics & Clinical on PET Images (Extra-nodal) |
|||
|
|
Variables |
Hazard Ratio |
P-value |
|
|
MTV |
1.183 [1.064,1.315] |
0.00184* |
|
|
LDH |
1.00 [0.999,1.001] |
0.7837 |
|
|
Texture PC1 |
0.968 [0.898,1.045] |
0.40542 |
|
|
Shape PC1 |
1.218 [1.048,1.416] |
0.01011* |
|
|
Texture PC1: Shape PC1 |
0.986 [0.949,1.025] |
0.48749 |
Reference:
Mikhaeel, G., et al., J of Clinc Onco, vol 40 (21), 2022; https://doi.org/10.1200/JCO.21.0206
Dean et al., Blood Adv, 2020;4(14): doi: 10.1182/bloodadvances.2020001900.
Balagurunathan et al., Front Oncol. 2024 vol.25(14). doi: 10.3389/fonc.2024.1485039.
G Iacoboni et al., Hemasphere 2024 May 21;8(5):e62. doi: 10.1002/hem3.62
H Schöder, et al., J Clin Oncol. 2016; 34(30), 10.1200/JCO.2016.69.3747; PMID: 27601547
R#1-Q#3
3- Radiomic features are extracted at the lesion level, but survival outcomes (OS/PFS) are assessed at the patient level, leading to potential statistical inconsistency and oversimplification.
Response to R#1-Q#3. We understand the concern. In radiological assessment, the largest lesion in a patient is often considered a representative of the disease condition. Which follows most radiological assessments to report the disease condition, including the disease progression (see RECIST criteria for solid lesions or RECIL criteria for non-solid tumors).
In lymphoma, the Deauville Scale assessment is reported for the focal lesions, which are the most avid lesions (limited to 6). Our study direction follow the direction of current clinical assessment.
PET/RECIST:
Wahl et. Al., J Nucl Med, 2009; doi: 10.2967/jnumed.108.057307.
Schwart, et al., Eur J Cancer, 2016; 62., doi: 10.1016/j.ejca.2016.03.081.
Berz., et al., Front Oncol, 2022:12; doi: 10.3389/fonc.2022.98298
N Papathanasiou, Hell J Nucl Med, 2023 (PMID: 37658560)
R#1-Q#4
4- While radiomic PCs and MTV were compared descriptively, no formal statistical comparison of predictive models (e.g. ΔC-index, NRI, IDI) was performed.
Response to R#1-Q#4:
We understand the concern. The study’s goal is to evaluate ability of radiomic based metrics to prognosticate treatment outcome and assess the outcome differences quantified using the logrank test. We computed c-index for harzard rations and compared with the clinical reference (MTV) for the same cohort. In comparison, most clinical scores such as the NRI (see below) had smaller dynamic range (liker scale), also see our response to R1-Q2.
Review of the metric:
Nomogram revised index (NRI) : Which includes clinical metrics and two levels.
1 point each for the risk factor age >60 years, ECOG score ≥2, elevated LDH, PTI, or stage II; 2 points for stage III/IV disease.
IPI Index: 5- levels
Uses: age, stage LDH, ECOG, Extra nodal sites
R#1-Q#5
5- The manuscript provides a minimal descriptive account of baseline variables, which undermines the robustness of the downstream survival analysis. Given the central role of patient heterogeneity in DLBCL outcomes, a more comparative analysis of baseline data is essential.
Response to R#1-Q#5
We understand the concern. We provided a brief description of the baseline variables as part of patient collection (see below) and details deferred to prior publications (Front Oncol. 2024 vol.25(14). doi: 10.3389/fonc.2024.1485039 and Blood Adv . 2020 Jul 28;4(14):3268-3276. doi: 10.1182/bloodadvances.2020001900.). We added details to supplemental section.
“Patients with non-measurable lesions or with or without baseline PET imaging were excluded from the study. Bridging therapy was defined as any lymphoma specific therapy given before CAR T-cell infusion, prior to the start of fludarabine cyclophosphamide chemotherapy for lymphodepletion, previously presented [1,2]. “
We agree that DLBCL is a heterogeneous disease, and patient collection adds to the problem. We followed clinical criteria for patient eligibility for the CAR-T trial and enrolled patients for the study with adequate follow-up.
Reviewer 2 Report
Comments and Suggestions for Authors
In this study, we investigated the role of radiometrics (radiomics) in PET/CT images in predicting prognosis and clarified its association with response to immunotherapy. In this study, radiomics signatures based on shape and size features were identified as prognostic predictors of Axicel therapy, which is very significant. However, the following improvements should be noted. 1) In this study, we systematically clarified that the (PET) feature PC based on the shape of extranodal lesions is a prognostic factor for survival rate (OS/PFS) (1 year). I think it would be better to add a discussion on why only extranodal lesions showed a significant difference? Conversely, why nodal lesions did not show a significant difference? 2) In this study, the radiological features calculated across categories (size, shape, texture) reflect tumor volume and malignancy, so I think it would be better to add a discussion on how they are useful for predicting treatment outcomes after Axicel therapy. Please also discuss and explain the relationship with metabolic tumor volume (MTV). I think adding a discussion of the relationship with tumor SUVs would improve understanding.
Author Response
Reviewer #2: (R#2)
In this study, we investigated the role of radiometrics (radiomics) in PET/CT images in predicting prognosis and clarified its association with response to immunotherapy. In this study, radiomics signatures based on shape and size features were identified as prognostic predictors of Axicel therapy, which is very significant.
R#2, Q1: However, the following improvements should be noted. 1) In this study, we systematically clarified that the (PET) feature PC based on the shape of extranodal lesions is a prognostic factor for survival rate (OS/PFS) (1 year). I think it would be better to add a discussion on why only extranodal lesions showed a significant difference? Conversely, why nodal lesions did not show a significant difference?
Response to R#2, Q1: We understand the concern. In DLBCL, it has been reported that extranodal disease patients have been shown to have worse outcomes and have been adopted in the clinical risk categorization (see references below). Our study evaluates patients with the largest nodal and extra-nodal disease as seen on PET/CT imaging and agnostically relates the radiomic metrics to the patient’s treatment outcome. We find that radiomic imaging characteristics (Shape PC) on PET images of the extra-nodal cohort show a significant survival outcome (see Table 4). While the radiomic characteristics on CT images or the nodal disease cohorts show borderline significance. Our findings agnostically converge with the clinical hypothesis related to treatment outcomes.
References:
a.Response rates of extra-nodal diffuse large B cell lymphoma to anti-CD19-CAR T cells: A real word retrospective multicenter study, Eur J Haematol, 2023 Jul;111(1):63-71. doi: 10.1111/ejh.13968.
c.Primary extra-nodal diffuse large B-cell lymphoma: A prognostic analysis of 141 patients, 2018, Onco Lett. doi: 10.3892/ol.2018.8803
Blood-Supplemental (ASH proceedings)
- Clinical characteristics and outcomes of extranodal stage I diffuse large B-cell lymphoma in the rituximab era, Blood 2021. 10.1182/blood.2020005112
-Response Rates of Extra-Nodal Diffuse Large B Cell Lymphoma to CD19-CAR T Cells - a Real Word Retrospective Multi-Center Study, Blood Suppl. 2022, https://doi.org/10.1182/blood-2022-165149
-Extra Nodal Diffuse Large B Cell Lymphoma: Site Specific Survival Study, ASH conference 2021, https://doi.org/10.1182/blood-2021-149328
-. A New Simplified Prognostic Index Integrating the Type of Extra-Nodal Involvement and Age for Ann Arbor Stage IV Hodgkin Lymphoma Patients Diagnosed at TEP-Scanner Era: A Retrospective Analysis from Lymphoma Study Association (LYSA) Centers, Blood Suppl, 2018, https://doi.org/10.1182/blood-2018-99-118578
R#2, Q2:
2) In this study, the radiological features calculated across categories (size, shape, texture) reflect tumor volume and malignancy, so I think it would be better to add a discussion on how they are useful for predicting treatment outcomes after Axicel therapy. Please also discuss and explain the relationship with metabolic tumor volume (MTV). I think adding a discussion of the relationship with tumor SUVs would improve understanding.
Response to R#2, Q2: We understand the concern. We updated the discussion section to highlight the differences between the feature categories and their relation to the MTV.
“In comparison, our study identifies the largest lesion characteristics, across three major descriptive categories (size, shape, texture), and subdivided based on nodal status (nodal, extranodal) to relate them to treatment outcome. We have shown shpe based descriptors in PET imaging among the patients in the extranodal cohort shows significant treatment outcome differences (OS/PFS, 1-year).”
“It has been reported that most proliferative lesions are non-circular, and irregular compared to benign lesions[3,4]. A recent publication [5] reported that a high metabolic heterogeneity measured as a cumulative SUV-histogram on PET images has a poor prognosis for patients with DLBCL. “
Reviewer 3 Report
Comments and Suggestions for Authors
This study presents a comprehensive and innovative analysis of radiomic features derived from PET/CT imaging to prognosticate treatment response in patients with relapsed/refractory diffuse large B-cell lymphoma undergoing CAR-T cell therapy. Below are specific recommendations to address these aspects.
- The majority of the cited references are over five years old and do not adequately reflect recent advancements in the field. It is recommended to incorporate high-quality literature from the past five years to ensure the study’s relevance and alignment with current research trends.
- The study included 155 patients. Was a sample size calculation performed? If not, please provide a power analysis to justify the adequacy of this sample size and discuss the potential implications of any uncertainties in the results.
- In line 139, principal components (PCs) are derived through linear transformation (PCA), which inherently assumes linearity. However, the Spearman correlation coefficient is designed to assess monotonic nonlinear relationships. This may represent a methodological mismatch. Please elaborate on the rationale for choosing the Spearman correlation coefficient in this context.
- The figures and tables require refinement to enhance their clarity and professionalism. Please optimize their layout, formatting, and overall visual appeal to improve readability and presentation quality.
- The manuscript would benefit from thorough English language editing by professional editing service to address grammatical errors, typos, and improve overall fluency.
- While the conclusions focus on CAR-T therapy for DLBCL, the discussion does not explore the potential applicability of radiomic features to other lymphoma subtypes or solid tumors. It is recommended to expand the discussion to include these aspects, highlighting the broader relevance of the findings.
- The study suggests that radiomic features can complement existing biomarkers. It would be valuable to analyze whether the predictive performance of a combined model (integrating radiomics with existing biomarkers) is significantly superior to that of individual indicators alone.
- The discussion could further explore the clinical translation pathway of the radiomic model, such as how to integrate it into current treatment decision-making workflows or what prospective validation steps are needed to achieve clinical application.
- The study does not address the potential association between radiomic features and CAR-T therapy-specific toxicities (e.g., CRS or ICANS). It is recommended to discuss this direction in the limitations section as an implication for future research.
Author Response
Reviewer#3:
Comments and Suggestions for Authors
This study presents a comprehensive and innovative analysis of radiomic features derived from PET/CT imaging to prognosticate treatment response in patients with relapsed/refractory diffuse large B-cell lymphoma undergoing CAR-T cell therapy. Below are specific recommendations to address these aspects.
R#3-Q#1:
- The majority of the cited references are over five years old and do not adequately reflect recent advancements in the field. It is recommended to incorporate high-quality literature from the past five years to ensure the study’s relevance and alignment with current research trends.
Response to R#3-Q#1: We understand the concern. We updated the reference to reflect current developments in the field. We have retained older references related to disease and study motivation.
Updated references:
Related to quantitative imaging:
Ligero, M.; Simó, M.; Carpio, C.; Iacoboni, G.; Balaguer-Montero, M.; Navarro, V.; Sánchez-Salinas, M.A.; Bobillo, S.; Marín-Niebla, A.; Iraola-Truchuelo, J.; et al. PET-based radiomics signature can predict durable responses to CAR T-cell therapy in patients with large B-cell lymphoma. EJHaem 2023, 4, 1081-1088, doi:10.1002/jha2.757.
Y Balagurunathan; Z Wei; J Qi; Z Thompson; E Dean; H Lu; S Vardhanabhuti; S Corallo.; J W Choi; J J Kim; et al. Ra-diomic Features on PET/CT Imaging of Large B cell Lymphoma Lesions Predicts CAR T-cell Therapy Efficacy. Frontiers in Oncology, section Hematological Malignancies 2024, Volume 14 - 2024, doi:https://doi.org/10.3389/fonc.2024.1485039.
DLBCL
Senjo, H.; Hirata, K.; Izumiyama, K.; Minauchi, K.; Tsukamoto, E.; Itoh, K.; Kanaya, M.; Mori, A.; Ota, S.; Hashimoto, D.; et al. High metabolic heterogeneity on baseline 18FDG-PET/CT scan as a poor prognostic factor for newly diagnosed diffuse large B-cell lymphoma. Blood Adv 2020, 4, 2286-2296, doi:10.1182/bloodadvances.2020001816.
R#3-Q#2:
- The study included 155 patients. Was a sample size calculation performed? If not, please provide a power analysis to justify the adequacy of this sample size and discuss the potential implications of any uncertainties in the results.
Response to R#3-Q#2:
We understand the concern. Our study has a patient sample count necessary for the proposed study. Most statistical sample size analysis related to prognosis would require assumptions that include event rate, extent of follow up between the groups.
In this pilot study, we followed a rough calculation to curate patient records that would be needed to power the analysis ( 80% power with acceptable false positive of 5%), with a follow-up time of 12 months and an expected longer spread in the patient follow- up time across the patient population between the survival arms (ratio of, 1:2.3). We find that a minimum of 124 patients would be necessary to achieve the required power.
R#3-Q#3:
- In line 139, principal components (PCs) are derived through linear transformation (PCA), which inherently assumes linearity. However, the Spearman correlation coefficient is designed to assess monotonic nonlinear relationships. This may represent a methodological mismatch. Please elaborate on the rationale for choosing the Spearman correlation coefficient in this context.
Response to R#3-Q#3: We understand the concern. The imaging metrics used in this work are a combination of linear and non-linear characterization of an abnormal region seen on the imaging scan (PET or CT). A PCA component is a linear combination of features that make up the orthogonal space (different weighting on the features, some may not contribute), the approach followed in this work is certainly valid.
Spearman’s correlation tests a monotonic relationship between two variables without scaling, remapping, or standardization. Which is valid to compare the trend of these new variables with known clinical metrics (metabolic tumor volume).
R#3-Q#4:
- The figures and tables require refinement to enhance their clarity and professionalism. Please optimize their layout, formatting, and overall visual appeal to improve readability and presentation quality.
Response to R#3-Q#4:
We understand the concern. We will work with the Cancer Journal to ensure that the figures and tables comply with the type set and formatting. The current version has a few formatting issues related to the tables that are not presenting well. We will correct them in the next version and update any missed out changes.
R#3-Q#5:
- The manuscript would benefit from thorough English language editing by professional editing service to address grammatical errors, typos, and improve overall fluency.
Response to R#3-Q#5:
We understand the concern. The article was corrected for grammatical errors and proof read to improve presentation.
R#3-Q#6:
- While the conclusions focus on CAR-T therapy for DLBCL, the discussion does not explore the potential applicability of radiomic features to other lymphoma subtypes or solid tumors. It is recommended to expand the discussion to include these aspects, highlighting the broader relevance of the findings.
Response to R#3-Q#6:
We understand the concern. We have referred to prior publications that relate shape metrics in solid tumors, some of which are related to our findings. The proposed methodology in terms of feature extraction and analysis has been followed in different flavors in other diseases. DLBCL is unique due to imaging scans (PET/CT) and current clinical inference is based on PET imaging.
R#3-Q#7:
- The study suggests that radiomic features can complement existing biomarkers. It would be valuable to analyze whether the predictive performance of a combined model (integrating radiomics with existing biomarkers) is significantly superior to that of individual indicators alone.
Response to R#3-Q#7:
We understand the concern. Our prior study and others have followed the suggested approach.
R#3-Q#8:
- The discussion could further explore the clinical translation pathway of the radiomic model, such as how to integrate it into current treatment decision-making workflows or what prospective validation steps are needed to achieve clinical application.
Response to R#3-Q#8: We understand the concern. We have added a few suggested chages in the discussion section, related to clinical translations. We believe using these findings (image-based characteristics) in clinical practice will allow better patient enrollment, tailor treatment, and use them as a precursor for patient enrollment that would benefit patients affected by this disease.
R#3-Q#9:
- The study does not address the potential association between radiomic features and CAR-T therapy-specific toxicities (e.g., CRS or ICANS). It is recommended to discuss this direction in the limitations section as an implication for future research.
Response to R#3-Q#9: We understand the concern. The suggestions are valid and important questions for clinical care, but they are beyond the scope of the study goals. We added to the study's limitations.
References:
- Dean, E.A.; Mhaskar, R.S.; Lu, H.; Mousa, M.S.; Krivenko, G.S.; Lazaryan, A.; Bachmeier, C.A.; Chavez, J.C.; Nishihori, T.; Davila, M.L.; et al. High metabolic tumor volume is associated with decreased efficacy of axicabtagene ciloleucel in large B-cell lymphoma. Blood advances 2020, 4, 3268-3276, doi:10.1182/bloodadvances.2020001900.
- Y Balagurunathan; Z Wei; J Qi; Z Thompson; E Dean; H Lu; S Vardhanabhuti; S Corallo.; J W Choi; J J Kim; et al. Radiomic Features on PET/CT Imaging of Large B cell Lymphoma Lesions Predicts CAR T-cell Therapy Efficacy. Frontiers in Oncology, section Hematological Malignancies 2024, Volume 14 - 2024, doi:https://doi.org/10.3389/fonc.2024.1485039.
- Jalaguier-Coudray, A.; Thomassin-Piana, J. Solid masses: What are the underlying histopathological lesions? Diagnostic and Interventional Imaging 2014, 95, 153-168, doi:https://doi.org/10.1016/j.diii.2013.12.014.
- Zhang, J.; Li, Y.; Zhao, Y.; Qiao, J. CT and MRI of superficial solid tumors. Quant Imaging Med Surg 2018, 8, 232-251, doi:10.21037/qims.2018.03.03.
- Senjo, H.; Hirata, K.; Izumiyama, K.; Minauchi, K.; Tsukamoto, E.; Itoh, K.; Kanaya, M.; Mori, A.; Ota, S.; Hashimoto, D.; et al. High metabolic heterogeneity on baseline 18FDG-PET/CT scan as a poor prognostic factor for newly diagnosed diffuse large B-cell lymphoma. Blood advances 2020, 4, 2286-2296, doi:10.1182/bloodadvances.2020001816.
Round 2
Reviewer 1 Report
Comments and Suggestions for Authors
I am satisfied that the authors have addressed all of my previous concerns about the article. It is now much improved and I feel that it is now suitable for publication.
Author Response
Thank you
Reviewer 3 Report
Comments and Suggestions for Authors
- In the limitations section, the authors briefly mention CAR-T–associated toxicities. It would strengthen the manuscript to expand on this point by incorporating recent findings(https://doi.org/10.1182/blood-2022-165179ï¼›https://doi.org/10.1182/blood-2021-149050ï¼›https://doi.org/10.1182/blood-2023-183057).
- Given that this article is submitted to a special issue focusing on AI integration into clinical oncology PET/CT imaging, the authors may consider briefly discussing how artificial intelligence, including radiomics-based models, can be further leveraged to support treatment decision-making in CAR-T therapy(PMID: 40349154).
Author Response
R#1:
I am satisfied that the authors have addressed all of my previous concerns about the article. It is now much improved and I feel that it is now suitable for publication.
Response:
Thank you for your critical comments.
R#2:
- In the limitations section, the authors briefly mention CAR-T–associated toxicities. It would strengthen the manuscript to expand on this point by incorporating recent findings(https://doi.org/10.1182/blood-2022-165179ï¼›https://doi.org/10.1182/blood-2021-149050ï¼›https://doi.org/10.1182/blood-2023-183057).
Response: Thank you for the references. As noted, our study does not relate our findings to toxicities, and this was added as part of the limitation section.
We briefly mentioned the role of additional biomarkers to assess cytokines in biofluids to evaluate toxicities after CAR-T therapy.
The following statement was added:
“There are few studies that have tracked the toxicities related to advanced immunotherapy [1], which can certainly be an investigative area. Recent advancements in genomics biomarkers in other blood-based malignancies have shown promise in prognosticating disease progression [2,3] . “
- Given that this article is submitted to a special issue focusing on AI integration into clinical oncology PET/CT imaging, the authors may consider briefly discussing how artificial intelligence, including radiomics-based models, can be further leveraged to support treatment decision-making in CAR-T therapy(PMID: 40349154).
Response: Thank you for the reference. We have added the role of recent advancements in AI/Machine learning and their impacts on CAR-T (see prior response )
The following statement was added:
“Advancements in artificial intelligence methods will certainly play a role in identifying finer patterns in patients that could be used to improve patient care in these advanced treatments [4-6], there are several data requirements that need to be considered prior to clinical utility [7,8].”
Reference:
- Stella, F.; Chiappella, A.; Casadei, B.; Bramanti, S.; Ljevar, S.; Chiusolo, P.; Di Rocco, A.; Tisi, M.C.; Carrabba, M.G.; Cutini, I.; et al. A Multicenter Real-life Prospective Study of Axicabtagene Ciloleucel versus Tisagenlecleucel Toxicity and Outcomes in Large B-cell Lymphomas. Blood Cancer Discov 2024, 5, 318-330, doi:10.1158/2643-3230.Bcd-24-0052.
- Zhang, X.; Yang, X.; Wang, C.; Huang, L.; Zhang, Y.; Wei, J. High Expression of Plasma IL-1β Levels and Transition of Regulatory T-Cell Subsets Correlate with Disease Progression in Myelodysplastic Syndrome. Blood 2022, 140, 9761-9762, doi:10.1182/blood-2022-165179.
- Peng, J.; Yang, X.; Li, C.; Zhang, Y.; Ma, L.; Wei, J. High Serum ST2 Level Predicts Progression and Poor Prognosis of Denovomyelodysplastic Syndrome. Blood 2023, 142, 6455-6455, doi:10.1182/blood-2023-183057.
- Gong, Y.; Fei, P.; Zhang, Y.; Xu, Y.; Wei, J. From Multi-Omics to Visualization and Beyond: Bridging Micro and Macro Insights in CAR-T Cell Therapy. Adv Sci (Weinh) 2025, e2501095, doi:10.1002/advs.202501095.
- Luciani, F.; Safavi, A.; Guruprasad, P.; Chen, L.; Ruella, M. Advancing CAR T-cell Therapies with Artificial Intelligence: Opportunities and Challenges. Blood Cancer Discovery 2025, 6, 159-162, doi:10.1158/2643-3230.Bcd-23-0240.
- Shahzadi, M.; Rafique, H.; Waheed, A.; Naz, H.; Waheed, A.; Zokirova, F.R.; Khan, H. Artificial intelligence for chimeric antigen receptor-based therapies: a comprehensive review of current applications and future perspectives. Ther Adv Vaccines Immunother 2024, 12, 25151355241305856, doi:10.1177/25151355241305856.
- Balagurunathan, Y.; Mitchell, R.; El Naqa, I. Requirements and reliability of AI in the medical context. Physica Medica 2021, 83, 72-78, doi:https://doi.org/10.1016/j.ejmp.2021.02.024.
- Afroogh, S.; Akbari, A.; Malone, E.; Kargar, M.; Alambeigi, H. Trust in AI: progress, challenges, and future directions. Humanities and Social Sciences Communications 2024, 11, 1568, doi:10.1057/s41599-024-04044-8.